# Interactions between sensory prediction error and task error during implicit motor learning

**Jonathan S. Tsay** [1,2]*, **Adrian M. Haith** [3], **Richard B. Ivry** [1,2], **Hyosub E. Kim** [4,5]*

**1** Department of Psychology, University of California, Berkeley, California, United States of America, **2** Helen Wills Neuroscience Institute, University of California, Berkeley, California, United States of America, **3** Department of Neurology, Johns Hopkins University, Baltimore, Maryland, United States of America, **4** Department of Physical Therapy, University of Delaware, Newark, Delaware, United States of America, **5** Department of Psychological and Brain Sciences, University of Delaware, Newark, Delaware, United States of America

* xiaotsay2015@berkeley.edu (JST); hyosub@udel.edu (HEK)

**Data Availability Statement:** Data can be accessed at https://github.com/xiaotsay2015/target_jump.

**Funding:** RBI is funded by the NIH (NINDS: NS116883; NIDCD: DC0170941). HEK is funded by NIH K12 (HD055931) and NSF M3X 1934650. JST

## Abstract

Implicit motor recalibration allows us to flexibly move in novel and changing environments. Conventionally, implicit recalibration is thought to be driven by errors in predicting the sensory outcome of movement (i.e., sensory prediction errors). However, recent studies have shown that implicit recalibration is also influenced by errors in achieving the movement goal (i.e., task errors). Exactly how sensory prediction errors and task errors interact to drive implicit recalibration and, in particular, whether task errors alone might be sufficient to drive implicit recalibration remain unknown. To test this, we induced task errors in the absence of sensory prediction errors by displacing the target mid-movement. We found that task errors alone failed to induce implicit recalibration. In additional experiments, we simultaneously varied the size of sensory prediction errors and task errors. We found that implicit recalibration driven by sensory prediction errors could be continuously modulated by task errors, revealing an unappreciated dependency between these two sources of error. Moreover, implicit recalibration was attenuated when the target was simply flickered in its original location, even though this manipulation did not affect task error – an effect likely attributed to attention being directed away from the feedback cursor. Taken as a whole, the results were accounted for by a computational model in which sensory prediction errors and task errors, modulated by attention, interact to determine the extent of implicit recalibration.

## Author summary

What information does the brain use to maintain precise calibration of the sensorimotor system? Using a reaching task paired with computational modeling, we find that movements are implicitly recalibrated by errors in predicting both the sensory outcome of movement (i.e., sensory prediction errors) as well as errors in achieving the movement goal (i.e., task errors). Even though task errors alone do not elicit implicit recalibration, they nonetheless modulate implicit recalibration when sensory prediction errors are

is funded by the PODSII scholarship from the Foundation for Physical Therapy Research. The funders had no role in study design, data collection and analysis, decision to publish, or preparation of the manuscript.

**Competing interests:** The authors have declared that no competing interests exist.

present. The results elucidate an unappreciated interaction between these two sources of error in driving implicit recalibration.

## Introduction

Sensorimotor adaptation is an essential feature of human competence, allowing us to flexibly move in novel and changing environments [1–4]. Multiple learning processes have been shown to contribute to the performance changes observed in adaptation tasks, including an aiming process which is explicit, volitional, and learns rapidly and a recalibration process which is implicit, automatic, and learns slowly [5–10]. Recent work has focused on how these two learning processes may be driven by distinct error signals: Whereas explicit aiming responds to task error (TE), a signal reflecting task performance [6,11], implicit recalibration (a.k.a. implicit adaptation) responds to sensory prediction error (SPE), an error reflecting the difference between predicted and actual feedback [1,12–17]. Moreover, these two learning processes are thought to rely on distinct neural modules, with explicit aiming requiring more prefrontal control [18–20] and implicit recalibration requiring more cerebellar control [21–26].

However, recent results from visuomotor rotation tasks have motivated a broader perspective of implicit recalibration, and in particular, led to the proposal that implicit recalibration is sensitive not only to sensory prediction error, but also to task outcome. Empirically, the evidence supporting this hypothesis comes from studies in which perturbed visual feedback (the source of SPE) is combined with a manipulation of target size or target jumps [27–29] to create a condition in which the visual feedback "hits" the target (Fig 1). Adaptation in such situations is attenuated by about ~20% compared to that observed in control conditions with a similar SPE [30,31]. The hypothesis that implicit recalibration is sensitive to both SPE and task outcome is consistent with recent neurophysiological observations of reward-related activity in the cerebellum [32–36].

But how exactly are SPE and task outcome combined to drive implicit recalibration? One possibility is that behavior reflects the operation of two independent learning processes, one sensitive to SPE and the other sensitive to task outcome [30,31]. While this dual-error model is consistent with existing findings, it is unknown whether this reflects the operation of two learning processes that operate independently. For example, it remains to be seen if TE-only would be sufficient to drive adaptation, as would be predicted by such a dual-error model.

Alternatively, SPE and task outcome may interact. For example, the strength of the SPE might be modulated by task outcome; if the displaced cursor still manages to intersect the target, a reward signal linked with task success could weaken the system's sensitivity to SPE, reducing the rate of recalibration [37,38]. A different form of interaction might arise from processes tangential to recalibration. For example, displacement of the target, as is commonly used to manipulate TE, might capture attention and weaken the salience of the SPE. In principle, the interaction between TE and SPE could also be a combination of multiple effects.

To examine how SPE and TE collectively shape implicit recalibration, we performed a series of visuomotor experiments that systematically varied the size of these two errors. We also compared participants' performance to a series of computational models designed to catalogue potential ways in which SPE and TE may interact. To control the size of SPE (i.e., operationalized as the difference between the cursor feedback and the original target location), we used clamped visual feedback [13], in which the timing and extent of cursor motion is linked to hand motion, but the cursor trajectory is offset by a fixed angle relative to the target, and thus independent of the hand trajectory. To control the size of TE (i.e., operationalized as the

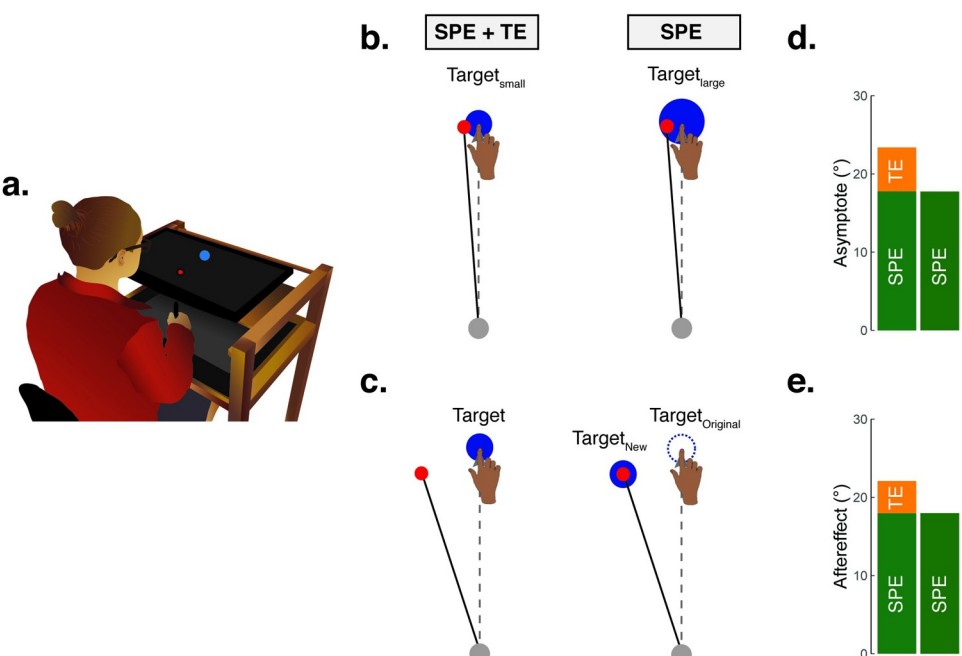

**Fig 1. Implicit recalibration elicited by SPE + TE and SPE-only.** (a) Illustration of experimental apparatus. **(b-c)** Task outcome was manipulated by either varying the size of the target (Kim et al, 2019) or varying the size of the target jump (Leow et al, 2018). Both SPE and TE are present when the cursor feedback straddles or misses the target, and only SPE is present when the cursor "hits" the target. **(d-e)** Implicit recalibration, as measured by the asymptote of hand angle in a clamped feedback design in Kim et al 2019 or during no-feedback aftereffect trials in a standard visuomotor rotation design, was reduced when TE was removed.

difference between the cursor feedback and the new target location), we jumped the target by a variable amount soon after movement initiation. In all cases, these manipulations were coupled with instructions to ignore the visual feedback and always reach straight towards the original target–an approach which has been shown to reliably elicit implicit recalibration without contamination from explicit strategies [30,39]. These experiments, coupled with computational models, allow us to precisely characterize the effects of SPE and TE on implicit recalibration.

## Results

### TE alone is not sufficient to drive implicit adaptation

We first examined whether TE-only perturbations would elicit implicit recalibration in Exp 1A (N = 12). The perturbation block was divided into four mini-blocks, each comprised of 201 trials with the same type of perturbation: SPE + TE, in which the cursor feedback was clamped between ±16˚ while the target remained stationary, or TE-only, in which the cursor feedback always moved through the original target (0˚ clamp) while the target jumped between ±16˚ away from its original position upon movement initiation. For SPE + TE trials, we expected the participant's movement would be shifted in the opposite direction of the cursor. For example, a leftward clamped cursor would elicit a rightward change in hand angle on the subsequent trial (Fig 2A). If TE alone is sufficient to elicit implicit recalibration, the participant's movement would be expected to shift in the direction of the jumped target on the subsequent trial. For example, a rightward target jump would be expected to elicit a rightward change in hand angle on the subsequent trial (Fig 2B).

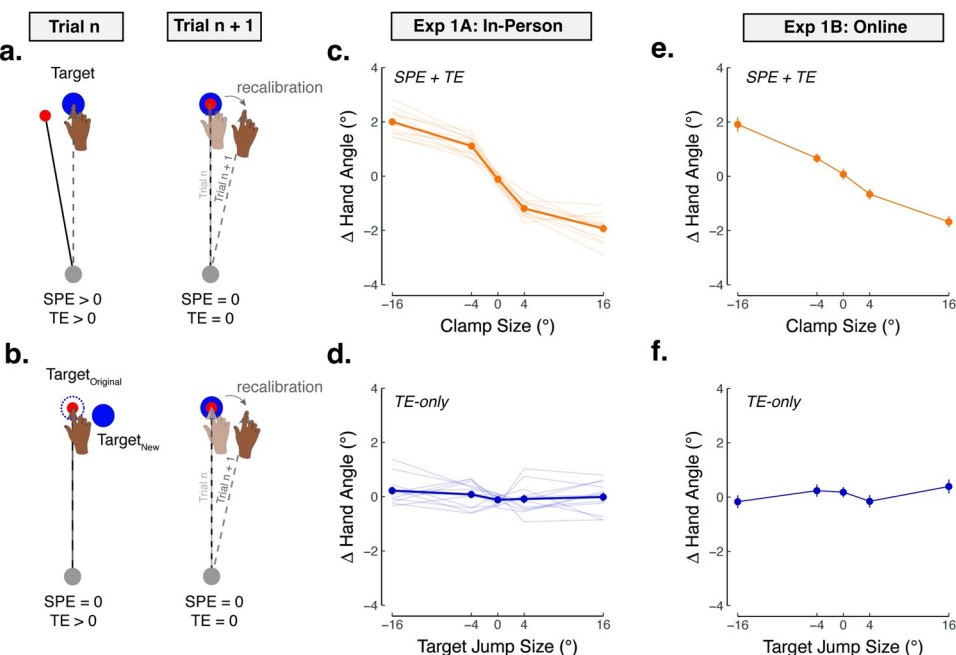

**Fig 2. Task error alone does not elicit implicit recalibration (Exp 1).** Using clamped visual feedback for testing implicit responses to: **(a)** SPE + TE, induced by offsetting the cursor trajectory at a fixed angle relative to the target (independent of the participant's heading angle), and **(b)** task error (TE) only, induced by jumping the target immediately after movement initiation, with the cursor clamped to 0˚ (the original target location; Exp 1A) or with no cursor feedback provided (Exp 1B). Note that for ease of exposition, we illustrated trial n + 1 as a TE-only trial (0˚ clamp paired with a 0˚ target jump); however, trial n + 1 could be another trial type (e.g., 4˚ clamped feedback paired with a 0˚ target jump). **(c)–(d)** Participants tested in the lab (Exp 1A) experienced alternating blocks of target jumps and clamped feedback (201 trials/block). The perturbation sizes within a given block were randomized to prevent accumulated learning. Adaptation was quantified by measuring how much the hand angle changed on trial n + 1 in response to the perturbation on trial n. **(e)–(f)** Participants tested online (Exp 1B) experienced a fully randomized schedule of target jumps and clamped feedback trials. Dots connected with thick line represent the across participant average; thin lines represent individual data. Due to the large number of participants in Exp 1B (N = 87), individual participants are not shown for ease of viewing.

In trials when both SPE and TE were present, all participants exhibited robust changes in hand angle to (partially) counter the imposed error, a key signature of implicit recalibration (Fig 2C; Mean slope ± SEM: $\beta = -0.1\pm0.0$; $F_{(1,212)} = 136.0$, $p = 1.3\times10^{-24}$, $\eta^2 = 0.2$). The change in hand angle as a function of error size appeared to be sublinear, composed of a linear zone for smaller perturbations (0˚– 4˚) and a saturated region for larger perturbations (4˚– 16˚), consistent with previous reports of saturated learning across a wide range of error sizes [12,13,40–42].

A very different picture was observed in the TE-only blocks. Here participants exhibited no reliable change in hand angle in response to the TE (Fig 2D; $\beta = 0.0\pm0.0$; $t_{(212)} = 0.6$, $p = 0.69$, $D = 0.1$). Critically, there was a striking interaction between perturbation size and perturbation type ($\beta = 1.2\pm0.1$; $F_{(1,212)} = 61.1$, $p = 2.5\times10^{-13}$, $\eta^2 = 0.2$), where robust implicit recalibration was observed when both SPE + TE were present, but not when TE-only was provided.

We tested the generality of this dissociation in two additional experiments. In Exp 1B we tested if the absence of recalibration on TE-only trials might be due to the presence of the clamp that moved directly to the target (i.e., 0˚ clamped feedback). Perhaps this salient visual signal might have distracted attention from the target jump or negated an error signal associated with the target jump. To address this, we conducted an online experiment in which no cursor feedback was provided on the TE-only trials (Exp 1B). As such, there was no visual SPE.

The only visual information in the display was the target which was displaced at movement onset on target jump trials. Because the experiment was conducted online, we were able to increase the sample size (N = 87). Once again, we observed a dissociation in which SPE + TE trials elicited robust sign-dependent changes in hand angle (Fig 2E), whereas TE-only trials resulted in no detectable changes in hand angle (Fig 2F). Statistically, there was an interaction between perturbation size and perturbation type ($\beta = -0.1\pm0.0$; $F_{(1,438)} = 92.4$, $p<0.001$, $\eta^2 = 0.2$), reflecting a negative slope in the SPE + TE function (Mean slope $\pm$ SEM: $\beta = -0.1\pm0.0$; $F_{(1,438)} = 160.4$, $p<0.001$, $\eta^2 = 0.1$), and no slope in the TE-only function (Mean slope $\pm$ SEM: $\beta = 0.0\pm0.0$; $F_{(1,438)} = 0.9$, $p = 0.35$, $\eta^2 = 0.0$). We note that the slope of the TE-only function remained indistinguishable with 0 even when we restricted our analysis to target jumps between $\pm4°$, see S2 Table.

We next considered whether the failure to show recalibration on TE-only trials might be due to the mini-block structure used in Exp 1A. In Exp 2, SPE + TE and TE-only trials were presented in a random, interleaved manner over the entire experiment (see Table 1; N = 40). The key measure of implicit recalibration was again the change in hand angle from trial n to trial n + 1 as a function of the error experienced on trial n. Robust sign-dependent changes in hand angle were observed for all participants in the SPE + TE condition (Set A: $\beta = -0.4\pm0.0$; $F_{(1,196)} = 138.7$, $p = 1.5\times10^{-24}$, $\eta^2 = 0.2$; Set B: $\beta = -0.4\pm0.0$; $F_{(1,196)} = 128.9$, $p = 2.8\times10^{-23}$, $\eta^2 = 0.1$; Fig 3A and 3B). In contrast, TE-only trials again failed to elicit any sign-dependent changes in hand angle (Set A: $\beta = 0.0\pm0.1$; $t_{(196)} = 0.5$, $p = 0.62$, $D = 0.1$; Set B: $\beta = 0.0\pm0.1$; $t_{(196)} = -0.4$, $p = 0.72$, $D = 0.1$; Fig 3C and 3D). The interaction between perturbation type and size was replicated showing robust implicit recalibration when both SPE + TE were present, but not when TE-only was provided (Set A: $\beta = 0.4\pm0.0$; $F_{(1,196)} = 67.5$, $p = 2.9\times10^{-14}$, $\eta^2 = 0.2$; Set B: $\beta = 0.4\pm0.0$; $F_{(1,196)} = 93.3$, $p = 2.7\times10^{-18}$, $\eta^2 = 0.3$),

Together, the results of Experiments 1 and 2 indicate that TE alone is not sufficient to drive implicit recalibration. This stands in contrast to SPE, which leads to implicit recalibration whether or not TE is present [30,31,43]. Moreover, these results challenge the hypothesis that SPE and TE operate strictly in an independent manner.

**Table 1. Summary of experiments.**

| | N | Setting | Perturbation Conditions | | | |
|---|---|---|---|---|---|---|
| | | | Set | Clamp size (°) | Target jump (°) | Figure |
| Exp 1 | 12 | In-Person | A | 0, ±4, ±16 | 0 | 2c |
| | | | | 0 | 0, ±4, ±16 | 2d |
| | 87 | Online | B | 0, ±4, ±16 | 0 | 2e |
| | | | | No Feedback | 0, ±4, ±16 | 2f |
| Exp 2 | 40 | In-Person | A | -4 | 0, -4, -8 | 3a, 5a |
| | | | | +4 | 0, +4, +8 | |
| | | | | 0 | 0, ±4 | 3c |
| | | | B | ±4 | 0, ±8 | 3b, 5a |
| | | | | 0 | 0, ±8 | 3d, 5a |
| Exp 3 | 100 | Online | A | ±3 | ±3, 0, $0_{\text{jump-in-place}}$ | 5b |
| | | | B | ±7 | ±3, 0, $0_{\text{jump-in-place}}$ | 5c |
| Exp 4 | 210 | Online | A | +3 | -10, -3, 0,+3,+7,+10,+17 | 6b |
| | | | | -3 | +10, +3, 0, -3, -7, -10, -17 | |
| | | | B | +7 | -10, -3, 0, +3, +7, +10, +17 | |
| | | | | -7 | +10, +3, 0, -3, -7, -10, -17 | |
| | | | C | ±3 | ±0, ±10, ±17, ±30 | |
| | | | D | ±7 | ±0, ±10, ±17, ±30 | |

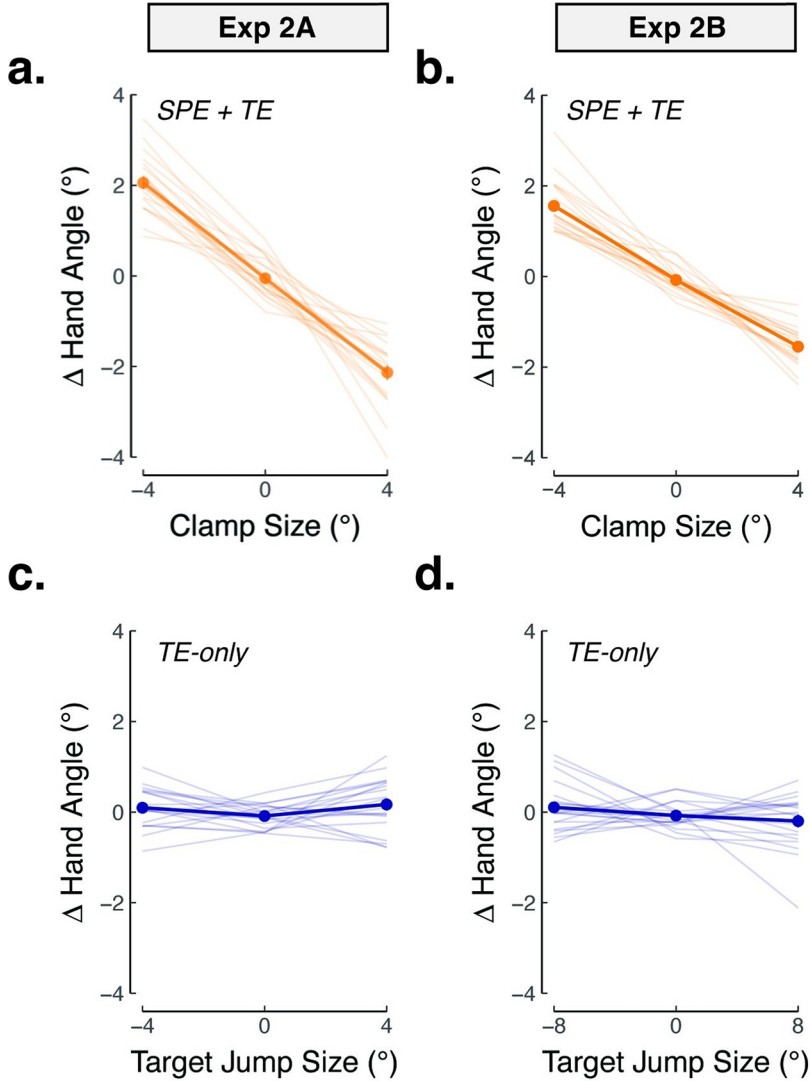

**Fig 3. Task error alone does not elicit implicit recalibration (Exp 2). (a)–(b)** Participants experienced a fully randomized (mixed) schedule of clamped feedback trials (Set A and Set B were both with ±4˚ SPE + TE perturbations) and **(c)–(d)** target jumps (Set A with ±4˚ TE-only perturbations and Set B with ±8˚ TE-only perturbations). Dots connected with thick line represent the across participant average; thin lines represent individual data.

## Modeling the potential ways in which TE and SPE may interact to drive implicit recalibration

Although TE alone may not induce recalibration, previous work has shown that the presence or absence of TE will modulate the response to SPE [30,31,43]. To understand the potential ways in which SPE and TE may interact to drive learning, we considered several models that encapsulated a variety of possible mechanisms. Fig 4 shows these models with their predicted responses to a fixed clamp size (i.e., fixed SPE) and varying TE size.

We first consider two simple base models, both of which cannot account for previously established results (including Experiment 1) but will serve as a foundation and a contrast for more elaborated models. The first model is one in which TE does not contribute to implicit recalibration. By this Invariant SPE model, we would expect recalibration to be invariant to

the size of target jumps and thus the size of TE (Fig 4A). As noted above, this model is insufficient given the demonstrations in the literature where the response to feedback involving only SPE is attenuated compared to feedback in which there is both SPE and TE (Fig 1).

The second base model is one in which TE and SPE make independent contributions to implicit recalibration (Dual-Error model), with their respective contributions simply being summed. Consequently, jumping the target in the same signed direction as the clamped cursor

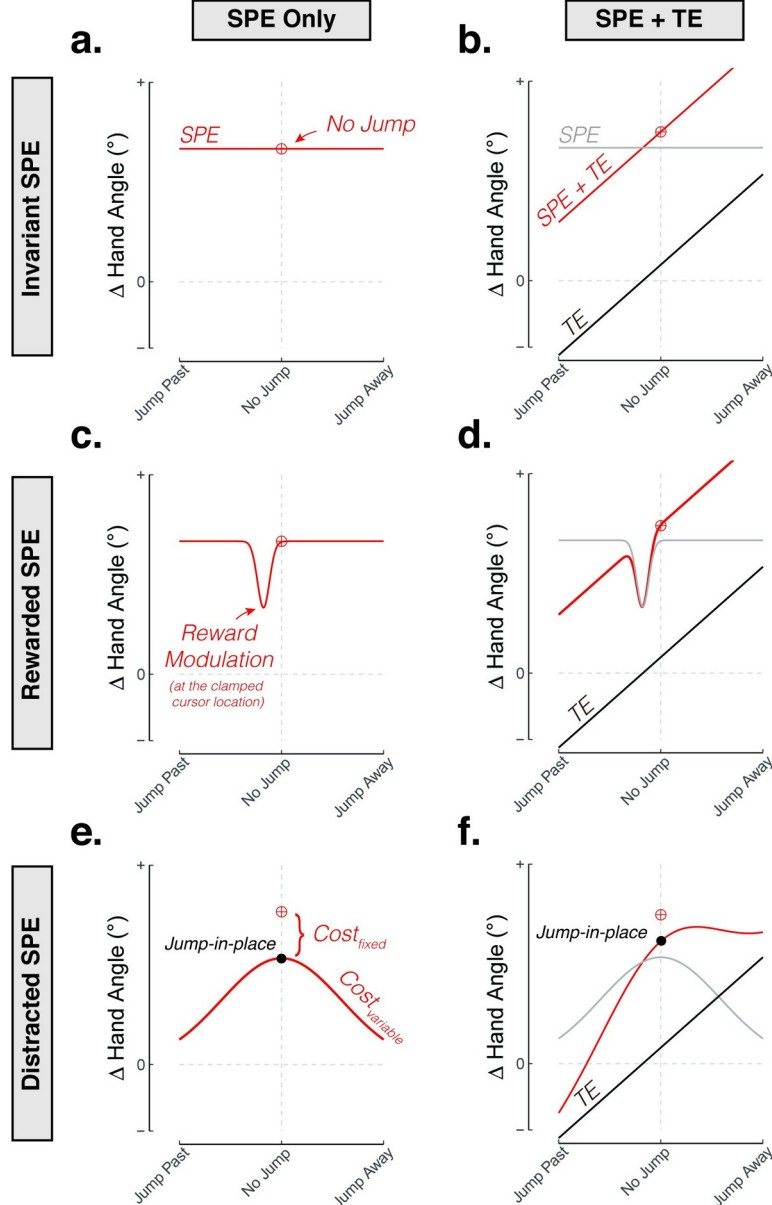

**Fig 4. Modeling the influence of target jumps on adaptation to TE and SPE.** Given a constant SPE magnitude, SPE may be (**a**) impervious to target jumps (**c**) attenuated when the cursor "hits" the (jumped) target (modulated by intrinsic reward), or (**e**) attenuated due to the motion of the jumping target diverting attention away from computing a SPE. The attenuation is assumed to be driven by the mere presence of a target jump (fixed cost–an effect isolated by flickering the target, also known as the jump-in-place condition) and varied with target jump size (variable cost). Right column (**b**), (**d**), (**f**): Adaptation may also be driven by a TE-based learning process, assumed here to be a linear function of the distance between the feedback and new position of the target. Note that TE is 0 at the clamped cursor location. The red indicates expected behavior, which is the composite of the SPE process (grey) and TE-based process (black).

(e.g., clockwise target jump and clockwise clamp) will decrease the absolute magnitude of TE. This ought to decrease recalibration since SPE and TE make opposing contributions to the behavioral change. Conversely, jumping the target away from the cursor will increase TE, and thus increase recalibration (Fig 4B). This model, however, cannot account for the failure of TE-only to elicit recalibration (see Figs 2D and 2F; 3C and 3D).

Building on the failure of these base models, we considered potential ways in which task outcome might influence recalibration to SPE in an interactive manner. One possible way is based on the hypothesis that recalibration is attenuated by a scalar intrinsic reward signal that simply indicates whether or not the movement goal was achieved (i.e., whether or not the cursor "hits" the target) [31,44–48]. The intrinsic reward signal can be interpreted as a gain controller, similar to previous efforts to model the effect of explicit rewards and punishments on recalibration [44]. That is, when the movement goal is achieved, the drive to recalibrate the motor system is reduced. This Rewarded SPE model predicts a transient drop in recalibration only for a narrow range of target jumps corresponding to the cursor hitting the target (Fig 4C).

An alternative model is that the target jump might be a distracting event, and this could result in an attenuated recalibration. Distraction could weaken the salience of the feedback signal [49,50] or increase visual uncertainty of the feedback signal [51–53], effects that have been hypothesized to weaken the error signal (but see: [51]). The target jump may also decrease the availability of the sensory prediction, a signal conveyed, at least in part, by the original target location [6]. Regardless of the exact mechanism, this Distracted SPE model is grounded in a rich history of visual psychophysics revealing worse accuracy at detecting, discriminating, and processing visual stimuli (feedback or target) in unattended regions of visual space [54,55]. Here, we simply assumed that displacing the target distracts attention away from the feedback cursor, and thus decreases the efficacy of recalibration. As a first approximation, we model this as a Gaussian gain function in which the attentional cost increases with the magnitude of the target jump (variable cost depicted in Fig 4E), an assumption we will test in Experiment 4.

This attentional hypothesis highlights that jumping the target has two effects: in addition to modifying the size of a putative TE signal, the standard motivation for this manipulation [30,43], it is a source of attentional distraction. One way to separate these factors is to transiently turn off the target while keeping its position fixed. Assuming the flicker serves to distract attention, this "jump-in-place" condition would identify an attentional cost that is independent of the change in TE, an assumption we will test in Experiment 3. This attenuating effect is shown in Fig 4E as a fixed attentional cost, that is, implicit recalibration when the target flickers in the same place during the trial (jump-in-place) would be attenuated compared to a condition when the target remains stationary and visible throughout the trial (no-jump). This fixed cost rides on top of a variable attentional cost that is dependent on the distance of the target displacement.

The Rewarded SPE (Fig 4C) and Distracted SPE (Fig 4E) models consider the modulatory effects of intrinsic reward and attention on a base model in which TE does not directly influence implicit recalibration (the Invariant SPE model). We also considered how the modulatory effects of reward and attention might influence implicit recalibration if both SPE and TE drive learning (Dual-Error model). The predictions of these hybrid, dual-error models are presented in Fig 4D (Rewarded SPE+TE) and 4F (Distracted SPE + TE), both of which predict an asymmetrical effect of target jumps.

## TE modulates implicit recalibration in the presence of SPE

To empirically examine the interactions between SPE and TE, and evaluate the models described above, we performed a second experiment in which we varied the size of target

jumps in the context of an SPE, induced by non-zero clamped feedback (see Table 1; N = 40). To vary the size of TE, we jumped the target between ±8˚ away from the original target location. For the non-zero SPE, we clamped the cursor at ±4˚ from the original target, randomizing the direction of the feedback cursor from trial to trial.

In response to a stationary target (i.e., no jump), participants adapted 1.5˚ in response to a 4˚ clamp (Fig 5A). When the target jumped towards the cursor, implicit recalibration was reduced in a roughly stepwise, linear manner (Table 2): Jump-to (i.e., target jumps to the cursor) reduced implicit recalibration by 13% and jump-past (i.e., target jumps in the direction of *and* beyond the cursor) reduced implicit recalibration by 33%. The fact that jumping the target influenced behavior argues against the Invariant SPE model; task outcome indeed influences behavior in the presence of SPE. The graded effect is also not compatible with the Rewarded SPE models (Fig 4C and 4D), as these models predict a modulating effect of target jumps only when the target intersects the cursor feedback, providing a putative intrinsic reward.

Implicit recalibration was greater when the target jumped away from the cursor compared to when it jumped past (0.3±0.1; $t_{(77)}$ = 2.6, $p$ = 0.01, $D$ = 0.8) (Fig 5A; Table 2). This pattern is most consistent with the unique, asymmetrical function predicted by the Distracted SPE + TE model (Fig 4E) and refutes the symmetrical function predicted by the Distracted SPE-only

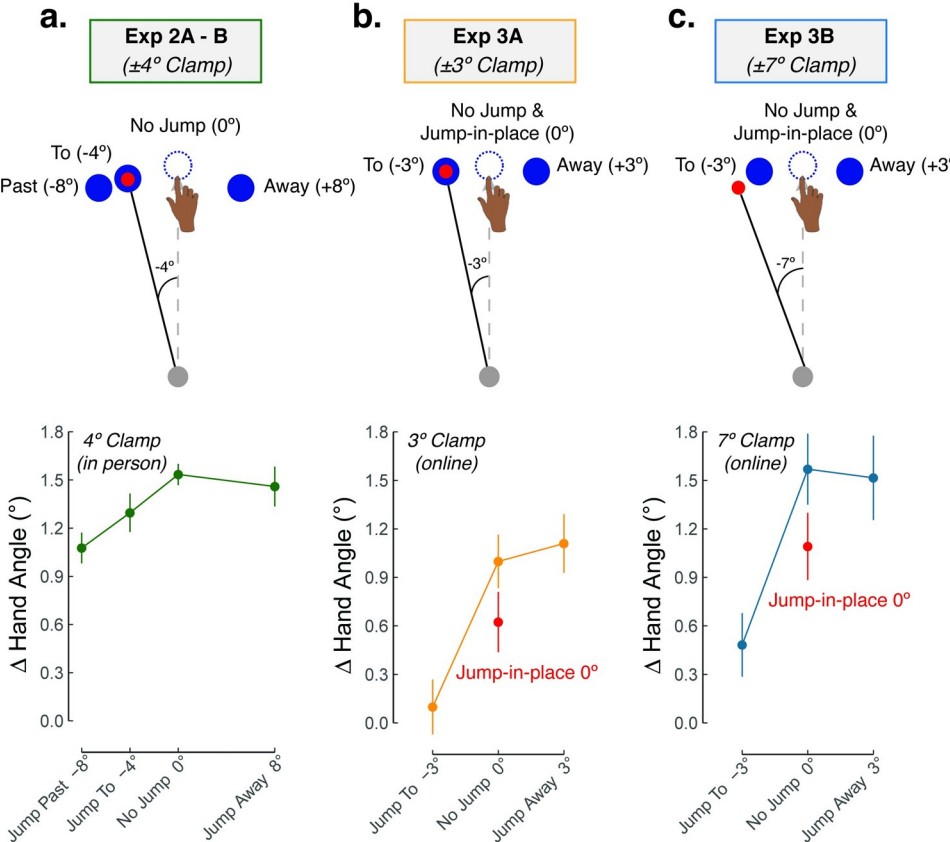

**Fig 5. Implicit recalibration is modulated by TE in the presence of SPE (Exp 2–3). (a)–(c)** Participants experienced a randomized zero-mean perturbation schedule where both clamp size (Exp 2, in-person: ±4˚ clamp; Exp 3, online: ±3˚ or ±7˚) and target jump size (Exp 2 range: -8 to 8; Exp 3 range: -3 to 3) were varied. A positive change in hand angle signified recalibration in the expected direction, by flipping the sign of hand angles in response to counterclockwise (+) clamped feedback and clockwise (-) target jumps. Dots represent mean and vertical lines represent SEM.

**Table 2. Summary of model-free results.**

| Fixed Effects | Exp 2A-B (-4° Clamp) | | | | Exp 3A (-3° Clamp) | | | | Exp 3B (-7° Clamp) | | | |
|---|---|---|---|---|---|---|---|---|---|---|---|---|
| | Past | To | No Jump | Away | To | Jump-in-place | No Jump | Away | Near | Jump-in-place | No Jump | Away |
| Target Jump Size | -8° | -4° | 0° | +8° | -3° | 0° | 0° | +3° | -3° | 0° | 0° | +3° |
| Mean (*SEM*) | 1.1 (0.1) | 1.3 (0.1) | 1.5 (0.1) | 1.4 (0.1) | 0.1 (0.2) | 0.6 (0.2) | 1.0 (0.2) | 1.1 (0.2) | 0.5 (0.2) | 1.1 (0.2) | 1.5 (0.2) | 1.5 (0.2) |
| Mean - No Jump (*SEM*) | -0.5 (0.1) | -0.2 (0.1) | | -0.1 (0.1) | -1.0 (0.2) | -0.4 (0.2) | | 0.0 (0.2) | -1.0 (0.2) | -0.4 (0.2) | | 0.0 (0.2) |
| D | -1.0 | -0.4 | | -0.2 | -0.8 | -0.3 | | 0.0 | 0.3 | 0.2 | | 0.1 |
| P | <0.001 | 0.14 | | 0.42 | <0.001 | 0.02 | | 0.88 | <0.001 | 0.02 | | 0.88 |

model (Fig 4F). That is, implicit recalibration may be dependent on both SPE and TE (conditioned on the presence of SPE), although the act of manipulating TE via target jumps may have a distracting effect that reduces sensitivity to SPE.

## Target jumps vary the size of TE but also attenuate implicit recalibration

Exp 3 was designed to provide a strong test of the assumption that jumping the target distracts attention: Namely we predict that recalibration in response to an SPE will be attenuated by distraction, even if the distracting event does not influence TE (or SPE). To test this prediction, we introduced a condition in which the target was perturbed *without* changing locations, disappearing upon movement initiation and then reappearing in its original location on the next screen refresh (jump-in-place; i.e., flickering the target). The difference between implicit recalibration for no-jump (i.e., stationary target) and jump-in-place should indicate the effect of distraction. By varying the size of the SPE, we can ask if the magnitude of the distraction effect is independent of SPE magnitude. To test this prediction, we used two clamp sizes (±3° and ±7°). This experiment was conducted online, making it readily amenable for inclusion of a large sample size (N = 100).

On average, participants adapted 1.1° and 1.5° in response to 3° and 7° clamps, respectively (no jump; Fig 5B and 5C; Table 2). Strikingly, the response was attenuated in the jump-in-place conditions even though the SPE and TE were identical to those experienced in the corresponding no-jump conditions. Moreover, the magnitude of this effect, which represents the fixed attentional cost on recalibration, was similar for the two clamp sizes, ~40% (no interaction: = 0.4±0.0; $F_{(3,294)}$ = 0.1, $p$ = 0.96, $\eta^2$ = 0). In addition to the fixed attentional cost due to the flicker of the target, we observed an approximately linear effect of TE on implicit recalibration. For instance, in Exp 3A recalibration was larger by approximately 0.5° in the jump-in-place condition compared to the jump-to condition, and increased by another 0.5° in the jump-away condition (Table 2). This linear effect of TE is uniquely predicted by the Distracted SPE + TE model.

In summary, the results indicate that perturbing the target yields 1) an asymmetrical hand angle function, 2) a fixed cost evident in the jump-in-place condition, and 3) a linear effect of TE *after* accounting for this fixed attentional cost. Based on these three effects, we can infer that implicit recalibration is impacted by the strength of TE, which itself is a function of the size of the target jump (Distracted SPE + TE model). Notably, these results were replicated in both the lab and online settings.

Mean estimates (SEM) from the linear mixed effect model for each target jump condition. Changes in hand angle in response to counterclockwise (+) clamped feedback were flipped to clockwise (-), such that a positive change in hand angle always signify adaptation in the expected direction (i.e., away from the clamped feedback). Contrasts between no jump and

other target jump conditions are also shown, with Cohens' D and P values provided. Significant contrasts ($P < 0.05$) are highlighted in a shaded light-grey box.

## Implicit recalibration reflects the joint contribution of TE, SPE, and the distractive effects of target jumps

To further probe how the distracting effect of target jumps interacts with the magnitude of TE, we sampled a wide range of target jump sizes in Experiment 4 (Fig 6A; N = 210). As shown in Fig 4F, we assume that the attenuating effect of distraction will increase with the size of the target jump due to attention being further displaced from the feedback cursor. As such, the inclusion of a larger range of target jumps should produce a marked asymmetrical function.

This prediction was confirmed (Fig 6B): Implicit recalibration decreased when the target jumped towards the cursor and remained relatively invariant when the target jumped away from the cursor, even as far as 30˚ (jump-away). This phenomenon could be attributed to the contribution of a TE process that offsets the attentional costs of target jumps on a SPE-based implicit recalibration process.

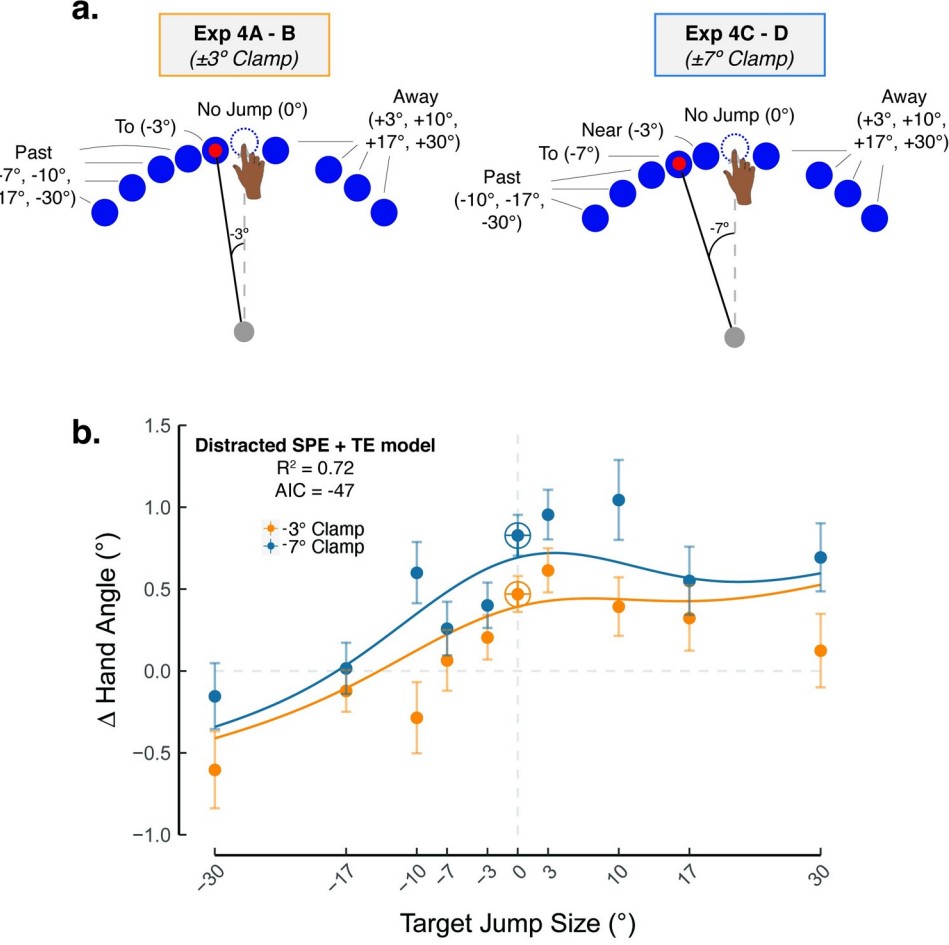

**Fig 6. Implicit recalibration reflects the contribution of learning from task error and sensory prediction error, with the latter sensitive to distraction from target jumps (Exp 4). . (a)–(b)** Participants experienced a randomized zero-mean perturbation schedule with clamp sizes (-3˚ shown in orange; -7˚ shown in blue) × target jumps (x axis, -30˚ through 30˚). The Distracted SPE + TE model was the winning model.

**Table 3. Summary of models.**

| Table 3 | SPE Only | SPE + TE |
|---|---|---|
| Invariant SPE | $U_{SPE} = U_{\theta_j=0}$ | $U_{Total} = U_{TE} + U_{SPE}$ $U_{TE} = \beta_{TE}(\theta_c - \theta_j)$ |
| Rewarded SPE | $U_{SPE} = U_{\theta_j=0} - \gamma_r e^{-(\theta_c - \theta_j)^2 / 2\sigma_r^2}$ | |
| Distracted SPE | $U_{SPE} = (C_J U_{\theta_j=0}) e^{(\theta_j)^2 / 2\sigma_d^2}$ | |

Parameters could either be free (red) or fixed (black, based on empirical data in Exp 4).

Sampling a wider range of target jumps also allowed us to fit our candidate models to the data (see formalization in Table 3 of the Materials and Methods section). In doing so, we could quantitatively evaluate how well our six candidate models fit the data while considering model complexity. Consistent with the qualitative assessments described above, the Distracted SPE + TE model provided the best fit, having the highest $R^2$ and lowest AIC (Table 4).

The modeling work also allowed us to evaluate the best fitting parameters of the Distracted SPE + TE model. The parameter values suggest that TE may contribute to learning. Given the estimated slope ($\beta_{TE}$) of the TE function was 0.02±0.003, we can infer that, of the 0.5˚ change in hand angle observed for the 3˚ no-jump condition (where both SPE and TE are present), 12.0% ± 1.8% of the change came from TE. Similarly, when the error increased to 7˚ in a no-jump condition, 15.5% ± 2.8% of the 0.9˚ change in hand angle came from TE. Importantly, these results indicate that SPE has a much larger impact on implicit recalibration than TE.

The Distracted SPE + TE model has two parameters to capture the effects of perturbing the target. First there is a fixed effect arising from the transient changes that occur when the target is perturbed. The estimate of this parameter ($C_J$) in the best fitting model was 0.84±0.13. Thus, the mere perturbation of the target, even if it was not spatially displaced reduced recalibration by 16.4% ± 13.0%. Second there is a variable cost ($\sigma_d^2$) due to SPE-based learning being attenuated as the target jump distance increased. The estimate of this parameter was 11.8±2.3. From this value, SPE would no longer be effective in driving implicit recalibration for target jumps greater than 35.5˚ ± 6.8˚ (i.e., $3\sigma_d^2$).

## Discussion

Although it is widely recognized that implicit sensorimotor recalibration serves to minimize motor execution errors, the error signals that drive this learning process remain the subject of considerable debate [1–3]. In particular, the idea that sensory prediction error (SPE), the mismatch between the expected and actual feedback, is the sole learning signal has been challenged by recent evidence demonstrating that task error (TE), the mismatch between the target location and feedback may also impact implicit recalibration [30,31,56,57]. Whether these two types of error drive implicit recalibration independently or interactively remains unknown.

**Table 4. Summary of model-based results.**

| | SPE Only | | | SPE + TE | | |
|---|---|---|---|---|---|---|
| | # of free param | $R^2$ | AIC | # of free param | $R^2$ | AIC |
| Invariant SPE | 0 | -0.54 | -18 | 1 | -0.54 | -18 |
| Rewarded SPE | 2 | 0.15 | -28 | 3 | 0.53 | -37 |
| Distracted SPE | 2 | 0.33 | -33 | 3 | 0.72 | -47 |

In traditional sensorimotor adaptation tasks, SPE and TE are confounded. Displacing the hand in a force field or perturbing the feedback in a visuomotor rotation task introduces both SPE and TE. To unconfound these signals, researchers have developed methods that selectively influence one signal or the other. For example, by making the angular trajectory of the feedback cursor independent of the movement, an SPE of a fixed size may either be accompanied by TE (when the target is small, and the cursor misses the target) or occur without TE (when the target is large, and the cursor hits the target). Conversely, displacing the target (i.e., target jump) selectively modulates TE given the assumption that the expected location of the feedback remains at the original target location.

Building on these methodological advances, we designed a series of experiments to systematically manipulate SPE and TE and used the data to test a set of computational models. We first considered a model in which these two types of error make independent contributions to implicit recalibration, with the resultant behavior being the composite operation of two distinct learning processes (Fig 4A and 4B). This idea takes inspiration from the work of Mazzoni and Krakauer (2006) who showed that implicit recalibration continued to operate even in the absence of task error, a result that suggests SPE-dependent learning is modular. A natural extension of this modular, dual-error model would posit that TE alone should also be sufficient to drive implicit recalibration. In four experiments, we failed to find support for this hypothesis. When the clamped feedback moved directly to the original target location (no SPE), hand angle remained unchanged in response to target jumps. That is, TE in the absence of SPE failed to induce implicit recalibration, arguing against models in which SPE-dependent and TE-dependent learning processes operate in a strictly independent manner.

Given the failure of this simple model and the dependency of TE on SPE, we considered different ways in which SPE-dependent and TE-dependent processes might interact. We varied task outcome in a continuous manner by jumping the target, either away from the perturbed cursor (increasing TE), towards the perturbed cursor (reducing TE), or to the location of the perturbed cursor (nullifying TE; i.e., SPE only). Whereas TE in the absence of SPE failed to elicit implicit recalibration, SPE in the absence of TE reliably elicited implicit recalibration. These observations obtained here using a trial-by-trial design are consistent with previous studies that have used a blocked design, in which SPE-only trials [30,31] or TE-only trials [58,59] remained invariant for an entire block.

By modulating TE in a fine-grained, continuous manner, we revealed an unexpected, asymmetrical effect on implicit recalibration: Implicit recalibration decreased when TE decreased yet remained largely unaffected when TE increased. These results are at odds with the hypothesis that task outcome provides a *binary* reward signal (Fig 4C and 4D), with TE being present when the cursor misses the target and TE being absent when the cursor hits the target [31,44–47]. This asymmetrical function is also at odds with the hypothesis where SPE-learning, the sole process driving implicit recalibration, is attenuated by a generic symmetric distractor effect of displacing the target (Fig 4E).

Instead, the pattern of results supports a hybrid model, where implicit recalibration is driven by both TE and SPE, with each error signal having a modulatory effect on the other error signal (Fig 4F). Implicit recalibration scales with the size of TE, but only when SPE is also present. Implicit recalibration also scales with the size of SPE but is attenuated when the target is perturbed. We hypothesize that the modulation of SPE-based learning occurs because attention is directed away from the feedback cursor and towards the (displaced) target, an effect that increases with the size of the displacement. Taken together, this hybrid perspective underscores the rich, dynamic interplay between two distinct error signals that drive implicit recalibration in an interactive manner.

We recognize that at this stage of development, the models are largely descriptive, intended to provide a qualitative sense of the behavioral changes that would be expected given different

ways in which sensory prediction error and task error might interact. Future work will be required to develop stronger theoretical foundations and more rigorous experimental tests for the different assumptions underlying the models; for example, to ask if the signals follow normative principles such as optimal integration [52,60] or relevance estimation [40].

While implicit recalibration seems to scale with TE in the presence of SPE, it remains unclear why TE alone fails to elicit recalibration. We consider here two, non-mutually exclusive possibilities. First, SPE may serve as a gating signal, with recalibration only engaged in the presence of SPE; when the gate is open the adaptation system responds to all sources of error information. Second, the lack of an SPE may have allowed the adaptation system to correctly attribute the target jump to an external cause [40,61–63]. Task errors were task-irrelevant in our experiments, with participants instructed to "ignore the target displacement and reach directly to the original position of the target". In contrast, when the instructions emphasize that the participant should "try to hit the target with the cursor", the task errors are task-relevant. It may be that the adaptation system is sensitive to task-relevant information. This latter hypothesis could account for the results of Ranjan and Smith (2020) who observed implicit adaptation in response to task-relevant TEs [64].

Saccade adaptation can be viewed as another task in which TEs are introduced in the absence of SPEs. In the typical setup, a visual target is presented at some peripheral location and the participant is instructed to make a saccade to the target. During the saccade, the target is displaced (e.g., to a more eccentric position). Not only do participants make a secondary saccade to bring the eye to the target, but adaptation occurs with the gain of the saccade modified in response to the error [65,66]. Since the target is displaced during the saccade, participants are unaware of the target displacement. Note that in saccade adaptation, the task error is task-relevant; not only do the instructions emphasize that the participant should look at the target, but the corrective saccades are produced automatically. Interestingly, if the target displacement occurs after the end of the saccade (and thus participants are aware of the perturbation), adaptation is attenuated [27]. While there are various accounts of the differential effects of intra- vs post-saccade displacements, it is possible that the visible displacement provides a cue that the error can be attributed to a perturbation in the environment rather than a poorly calibrated sensorimotor system.

The current study also highlights an important methodological issue. Similar to the way error clamps have provided a tool to isolate implicit recalibration, target jumps have been viewed as a way to provide a "pure" manipulation of TE. However, our results show an attenuated effect on implicit recalibration from the transient effects associated with perturbing the target, a result made salient by the conditions in which the target briefly disappeared and then reappeared at its original location. The transient sensory events associated with a target jump or flash might siphon attention away from the visual feedback, thereby weakening the overall learning signal. Alternatively, a transient distraction may have increased the likelihood that visual feedback is mis-localized, thus attenuating the motor system's reliance on this uncertain feedback [51–53,67,68]. Regardless of the mechanism, our results underscore the importance of considering the distractive effect of a target jump manipulation and the consequences of distraction on implicit recalibration.

## Materials and methods

### Ethics statement

All participants gave written informed consent in accordance with policies approved by the UC Berkeley's Institutional Review Board (Protocol Number: 2016-02-8439) and University of Delaware's Institutional Review Board (Protocol Number: 1320924–10).

## Participants and apparatus

All participants were between 18–30 years old and right-handed, as determined by either the Edinburgh handedness inventory [69] or through self-report.

In-person participants (Exp 1A – 2): Undergraduate students were recruited from the University of Delaware community, receiving financial compensation for their participation at a rate of $10/hour. Participants were seated in front of a custom tabletop setup and held the handle of a robot manipulandum (KinArm: BKIN Technologies, sampling rate 200 Hz) that was positioned below a mirror. Visual feedback was projected by a monitor placed directly above onto the mirror, which occluded vision of the participant's hand during the experiment. Peripheral vision of the arm was minimized by extinguishing the room lights. Participants completed the task by moving the robot manipulandum, which was constrained to a horizontal 2D plane.

Online participants (Exp 1B, 3, and 4): Participants were recruited via Amazon Mechanical Turk or Prolific, receiving financial compensation for their participation at a rate of $8/hour. Participants used their own laptop computer to access a customized webpage [70] hosted on Google Firebase (sampling rate typically ~60 Hz) [71,72]. Recruitment was restricted to trackpad users to minimize variability from different response devices. Participants completed the task by swiping their index finger on the trackpad.

## Reaching task procedure

In-person procedure: Reaches were made from a start location to one target (90˚ location, straight ahead). The start location was indicated by a white ring (6 mm diameter) and the target by a blue circle (6 mm diameter), with the radial distance between the start location and target fixed at 10 cm. To initiate a trial, the robot arm moved the participant's hand to the start location. Visual feedback of the hand position was given via a cursor (white circle 3.5 mm diameter) only when the hand was within 1 cm of the start position. Once the hand remained within the start location for 500 ms, the target appeared, serving as a cue to indicate the location of the target and an imperative to initiate the reach. To discourage on-line corrections, participants were instructed to perform fast, 'shooting' movements through the target as soon as the target appeared.

Reaction time (RT) was defined as the time from initial target presentation to the start of movement (defined as when the hand first exceeded 5 cm/s for at least 50 milliseconds). Movement time (MT) was defined as the time between the start of movement and when the hand crossed the radial target distance of 10 centimeters. To ensure that participants moved at a fast speed that excluded online feedback corrections, the message "Too Slow" appeared on the screen at the end of the trial when MT was < 40 cm/s at peak velocity. We also presented the message "Too Fast" if MT was > 70 cm/s at peak velocity to ensure that participants did not make simple ballistic movements in the general direction of the target (this criterion was rarely exceeded). After completing the reach, the participant was instructed to keep the arm and shoulder relaxed as the robot moved the hand back to the starting position.

Online procedure: The reaching task was adapted for an online study. We did not obtain information concerning the monitors used by each participant; as such, we cannot specify the exact size of the stimuli. However, from our experience in subsequent studies, we assume that most online participants used a laptop computer. To provide a rough sense of the stimulation conditions, we assume that the typical monitor had a 13" screen with a width of 1366 pixels and height of 768 pixel [72]. The center position was indicated by a white circle (0.5 cm in diameter) and the target location was indicated by a blue circle (also 0.5 cm in diameter). To ensure that reaches remain in the trackpad, we reduced the radial distance of the target to 6 cm and positioned the target at the 45˚ target (upper right quadrant).

The participant made center-out planar movements by moving the computer cursor with her trackpad to a visual target. To initiate each trial, the participant moved their hand to the start location. Visual feedback of the hand position was given via a cursor (white circle 0.5 cm diameter) when the hand was within 1 cm of the start position. Once the hand remained within the start location for 500 ms, the target appeared, serving as a cue to indicate the location of the target and an imperative to initiate the reach. To discourage on-line feedback corrections, participants were instructed to perform fast, 'shooting' movements through the target as soon as the target appeared.

Note that discouraging on-line feedback corrections was especially important for our study. There have been many classic studies investigating how target displacements impact on-line feedback control [73,74]. In particular, this work has specified the constraints governing how an initial movement trajectory may (or may not) be modified subsequent to a target jump. Here, our focus is not on on-line feedback corrections but, rather on how the displacement of the target impacts processes involved in maintaining the calibration of the sensorimotor motor system for feedforward control. Put differently, we wanted to know whether task error induced by a target jump on the current trial would modify the movement on the *next* trial.

RT was defined as the time from initial target presentation to the start of movement (i.e., when the hand movement exceeded 1 cm from the start location). Due to the lower sampling rate of standard computer monitors compared to in-person setup, we opted to define RT in terms of movement distance (requiring fewer samples) rather than movement velocity (requiring more samples to adequately estimate). There were no constraints on RT. MT was defined as the time between the start of the movement and when the radial distance of the movement reached 6 cm. To ensure that the movements were made quickly, the computer displayed a "too slow" message if MT exceeded 300 ms. We did not include a "too fast" message since participants recruited online, based on our pilot results, err on the side of moving too slowly.

There were three types of cursor feedback trials used throughout the in-person and online experiments: On veridical feedback trials, the cursor corresponded to the position of the hand. On clamped feedback trials, the cursor followed an invariant path along a constant angle with respect to the target. The radial distance of the cursor, relative to the start position, was yoked to the participant's hand. In both types of feedback trials, the radial position of the cursor matched the radial position of the hand until the movement amplitude reached the radial distance of the target, at which point the cursor froze for 50 ms. On no-feedback trials, the cursor was blanked when the target appeared, and did not re-appear until the participant had completed the reach and returned to the start location for the next trial.

There were also target jump trials, where upon movement initiation (i.e., in-person: velocity > 5 cm/s; online: radial distance > 1 cm), the original target was blanked and immediately re-positioned at a new target location (i.e., one screen refresh between offset of original target to onset of new target; in-person: within 1 ms; online: <15 ms, accounting for the delay in the monitor system [71,72]). We varied the size of the target jump and categorized these based on the relative position of the new target location to the clamped cursor position: jump-past, jump-to, jump-near, jump-away, and jump-in-place. When the target jumps in the direction of the clamped cursor feedback, the size of the target jump could either be greater than (jump-past), equal to (jump-to), or less than (jump-near) the clamped angle. On jump-away trials, the target was repositioned in the direction opposite to the clamped feedback. On jump-in-place trials, the target disappeared upon movement initiation (1 refresh) and then reappeared (1 refresh) in the same (original) location (<30 ms, accounting for delays in the system). While jump-in-place has a longer interval between successive displays of the target compared to other target jump conditions, this interval ensured that jump-in-place trials elicited a detectable disturbance to the visual display, something that was obvious in the other target jump conditions.

### Experiment 1A and 2: In-person experiments

Reaching trials were performed to the 90˚ target (straight ahead). The experiment began with 100 baseline reaching trials with veridical feedback, provided to familiarize the participants with the reaching task. These trials were used to emphasize that movements should "shoot" through the target and demonstrate that the feedback and target would disappear soon after the movement amplitude exceeded the radial distance of the target.

The participant then completed a block of perturbation trials. Just before the start of this block, the error clamp and target jump manipulations were described to the participant, and she was told to ignore the cursor "feedback" as well as any change in the position of the target, always attempting to reach directly to the original target. To help the participant understand the task irrelevant nature of the clamped feedback and target jump, three demonstration trials were provided. The target appeared straight ahead at 90˚ and the participant was told to reach to the left (demo 1), to the right (demo 2), and backward (demo 3). The cursor moved in a straight line with a 45˚ offset from the original target in all three trials, and the target "jumped" upon movement initiation 0˚ (demo 1), 45˚ (demo 2), and 90˚ (demo 3) away from the original target.

In Exp 1A, the perturbation block (Table 1; 804 perturbation trials = 4 mini-blocks x 201 trials/mini-block) was composed of mini-blocks with either SPE + TE perturbations (i.e., when clamped feedback is paired with a stationary target) or TE-only perturbations (i.e., when a 0˚ clamp is paired with a target jump). We opted to keep these perturbation conditions separate to minimize any interference or generalization of learning from one trial type to another [75,76]. SPE + TE and TE-only mini-blocks were interleaved, with the order counterbalanced across individuals. Within each mini-block, there were five unique trial types (SPE + TE miniblock: 0˚, ±4˚, ±16˚ clamp paired with a 0˚ target jump; TE-only mini-block: 0˚ clamp paired with a 0˚, ±4˚, ±16˚ target jump; see Table 1). Each trial type was repeated 40 times (with one exception, there were 41 trials with 0˚ clamp and 0˚ target jump). The trial types were presented in a pseudo-randomized manner to ensure that the mean error (i.e., SPE + TE, or TE-only) was 0˚ every 20 trials. Across the entire experiment, there were 80 trials of each clamp size x target jump combination (84 trials in the 0˚ clamp, 0˚ target jump condition).

The perturbation block in Exp 2 was not composed of mini-blocks. Instead, TE and SPE + TE trials were randomized across the entire experiment (724 trials) to evaluate whether our results from Exp 1 would hold under another perturbation schedule. To prevent any systematic drifts in hand angle to one direction, the trials were scheduled in a pseudorandomized manner such that the mean error was 0˚ every 24 trials. To sample a wider range of clamp size x target jump combinations while keeping the experiment within 1 hour to minimize fatigue, participants experienced different sets of perturbations (Set A or Set B). In Set A, the target always jumped in the same direction as the error clamp, while in Set B, the target either jumped in the same or in the opposite direction of the error clamp (Table 1). There were 80 trials per clamp size x target jump combination (84 trials for the 0˚ clamp, 0˚ target jump condition).

### Experiment 1B, 3, and 4: Online experiments

Due to the onset of the pandemic, Exp 1B, 3, and 4 were conducted online. With this approach, we were able to increase our sample size in an efficient manner, providing greater power to detect subtle differences between target jump conditions. We used an motor learning platform (OnPoint) [70,77] and recruited participants using Amazon Mechanical Turk. Despite substantial differences between in-person and online sensorimotor learning experiments (e.g., inperson: dark room to occlude vision of the hand; online: full visibility of the hand for trackpad

users), we have found that the results obtained online are quite similar to those obtained in-person [78].

We made several additional changes to the experiment. We included "attention checks" to verify whether participants attended to the task. Specifically, during the inter-trial interval, participants occasionally were instructed to make an arbitrary response (e.g., "Press the letter "b" to proceed."). If participants failed the make the specified keypress, the experiment was terminated. These attention checks were randomly introduced within the first 50 trials of the experiment. We also included "instruction checks" after our three demo trials to assess whether participants understood the nature of the error clamp and target jump manipulations: "Identify the correct statement. Press 'a': I will aim away from the original target. I will ignore the white dot. Press 'b': I will aim directly towards the original target location and ignore the white dot." The experiment was terminated if participants failed to make an accurate keypress (i.e., "b").

The block design in Exp 1B began with a baseline block of 20 trials with veridical feedback followed by 200 perturbation trials. The perturbation involved either clamped feedback (SPE and TE) or target jump (TE-only) trials. Critically, cursor feedback was not provided on the target jump trials to minimize possible distracting effects from the cursor (i.e., that might attenuate TE-only implicit recalibration). There were 20 trials per condition, which were all randomized in a zero-mean manner throughout the experiment.

The block structure in Exp 3 and 4 were the same, composed of a baseline block with veridical feedback (28 trials) and a perturbation block with clamp feedback paired with target jumps (Exp 3: 120 trials; Exp 4: 252 trials). All perturbation conditions were randomized in a zero-mean manner throughout the experiment. The perturbation conditions were again divided into sets (See Table 1; Exp 3: Sets A—B; Exp 4: Sets A—D) to sample a wider range of clamp size x target jump combinations, while keeping the experiment within 1 hour. There were 30 trials per clamp size x target jump combination in Exp 3 and 18 trials per combination in Exp 4.

## Data analysis, model free

The primary dependent variable of reach performance was the hand angle, defined as the hand position relative to the target when the movement amplitude reached the target distance (i.e., angle between the lines connecting start position to target and start position to hand).

Outlier responses were defined as trials in which the hand angle deviated by more than 3 standard deviations from a moving 5-trial window. These outlier trials were excluded from further analysis, since behavior on these trials could reflect attentional lapses or anticipatory movements to another target location (average percent of trials removed per participant ± SD: Exp 1: 0.2 ± 0.2%; Exp 2: 0.1 ± 0.2%; Exp 3: 0.8% ± 0.8%; Exp 4: 1.1 ± 0.1%).

As a measure of trial-by-trial implicit recalibration, we evaluated each participant's median change in hand angle on trial n + 1, as a function of the perturbation condition (clamp size x target jump) on trial n (Δ Hand Angle). This trial-by-trial change in hand angle has been used in many studies as a measure of implicit recalibration (e.g., [40,53,79]).

We sought to determine whether SPE + TE and TE-only perturbations elicit robust sign-dependent changes in hand angle (Exp 1 and 2). Specifically, in the SPE + TE condition, we expect implicit recalibration to result in a change in hand angle in the opposite direction of the error clamp (e.g., a CW clamp eliciting a CCW change in hand angle). In contrast, in the TE-only condition, we expect implicit recalibration to be in the same direction as the target jump (e.g., a CW target jump eliciting a CW change in hand angle). To better visualize the difference between SPE + TE and TE-only conditions, the sign of the target jump was flipped, such that

the expected change in hand angle would also be in the opposite direction of the perturbation (i.e., a negative target jump would elicit a positive change in hand angle). Each participants' data were submitted to a linear regression with perturbation size (Exp 1: 0, ±4˚, ±16˚; xp 2, Set A: 0, ±4˚; Exp 2, Set B: 0, ±4˚, ±8˚) and perturbation type (clamp vs target jump) as main effects. The mean regression slopes ($\beta$) ± SEM across participants were provided.

To ask whether the effect of TE would be conditional on the presence of SPE, we submitted each participants' data in Exps 2 and 3 to a linear regression with target jump size and task set as main effects. Post-hoc contrasts were performed using two tailed t-tests, and P values were Bonferroni corrected. The mean regression values ($\beta$) ± SEM across participants were provided.

## Data analysis, model based

In this section, we formalize the six models justified in the Results section titled: "*Modeling the potential ways in which TE and SPE may interact to drive implicit recalibration.*" The development of these models was based on different assumptions about how the size of target jumps ($\theta_j$) and the size of the error clamp ($\theta_c$) impact the processing of SPE and TE.

The first set of models posit that the motor system responds only to SPE (Table 3: SPE only column): First, SPE may be impervious to target jumps (Invariant SPE), where motor updates are not affected by target jumps ($U_{\theta_j=0}$, or the motor update during no-jump). Second, SPE may be attenuated when the cursor lands in the target, modulated by intrinsic reward (Rewarded SPE). The amount of reward modulation could vary with $\gamma_r$, a gain value determining the amount of attenuation, and $\sigma_r$, the standard deviation of reward function determining the scope of attenuation. Third, SPE may be attenuated due to a distracting effect of target jumps, which may siphon attention away from processing feedback and/or the movement goal (Distracted SPE). The attenuation may be due to the presence of a target jump (a fixed cost, $C_J$) and the size of the target jump (variable cost, modeled as a gaussian decay with standard deviation $\sigma_d$).

We recognize that the distracted SPE hypothesis may take on a different form, where there may only be a fixed cost or only be a variable cost (or a different type of variable cost, like an inverted gaussian). However, these models fail to qualitatively capture our results, and therefore, we opted not to include these models in our formal analysis. We also recognize that, at present, we only consider how target jump impacts learning from SPE, whereas target jumps may also impact learning from TE.

Alternatively, implicit recalibration may also be driven by both SPE and TE-based learning processes (Table 4: SPE + TE). The contribution of TE was assumed to vary with the distance between the cursor feedback and the new target position in a linear fashion. $\beta_{TE}$ captures the slope of this function, and the $\theta_c-\theta_j$ term constrains implicit recalibration from TE to 0 when TE is 0 (i.e., when the target jumps onto the cursor feedback). This model assumes the net motor update ($U_{Total}$) to be the sum of a SPE-based learning process ($U_{SPE}$) and a TE based learning process ($U_{TE}$).

We evaluated the six models by simultaneously fitting group-averaged data for the ±3˚ and ±7˚ clamp groups in Exp 4. To quantify model performance, we compared $R^2$ (i.e., the sum of squared errors of the fitted model compared to the null model, which is the mean of all data points) and AIC (Akaike Information Criterion) scores. The winning model was the model with the largest $R^2$ and the smallest AIC. In order to calculate confidence intervals for the parameter estimates, we applied standard bootstrapping techniques, constructing group-averaged hand angle data 1000 times by randomly resampling with replacement from the pool of participants within each group. We started with 10 different initial sets of parameter values

and estimated parameter values that minimized the least squared error between the boot-strapped data and the model output.

## Supporting information

**S1 Text. Similarities and differences between Experiments 1 and 2.**
(DOCX)

**S2 Text. Similarities and differences between in-person and online experiments.**
(DOCX)

**S1 Table. A kinematic comparison across experiments.**
(DOCX)

**S2 Table. TE-only fails to elicit implicit recalibration in response to small target displacements.**
(DOCX)

## Acknowledgments

We thank Joie Tang for her assistance with data collection (Exp 1A).

## Author Contributions

**Conceptualization:** Jonathan S. Tsay, Adrian M. Haith, Richard B. Ivry, Hyosub E. Kim.

**Data curation:** Jonathan S. Tsay, Hyosub E. Kim.

**Formal analysis:** Jonathan S. Tsay, Hyosub E. Kim.

**Funding acquisition:** Richard B. Ivry, Hyosub E. Kim.

**Investigation:** Jonathan S. Tsay, Adrian M. Haith, Richard B. Ivry, Hyosub E. Kim.

**Methodology:** Jonathan S. Tsay, Adrian M. Haith, Hyosub E. Kim.

**Project administration:** Jonathan S. Tsay, Hyosub E. Kim.

**Resources:** Hyosub E. Kim.

**Supervision:** Richard B. Ivry, Hyosub E. Kim.

**Validation:** Adrian M. Haith.

**Visualization:** Adrian M. Haith, Hyosub E. Kim.

**Writing – original draft:** Jonathan S. Tsay, Richard B. Ivry, Hyosub E. Kim.

**Writing – review & editing:** Jonathan S. Tsay, Adrian M. Haith, Richard B. Ivry, Hyosub E. Kim.

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
