## [Decision Letter · Decision Letter 0]

30 Sep 2021

Dear Dr. Tsay,

Thank you very much for submitting your manuscript "Interactions between sensory prediction error and task error during implicit motor learning" for consideration at PLOS Computational Biology. As with all papers reviewed by the journal, your manuscript was reviewed by members of the editorial board and by several independent reviewers. The reviewers appreciated the attention to an important topic and the original experimental design. Based on the reviews, this is a borderline minor revision (with a major caveat) based on the reviewer comments we received raising concerns with the interpretation and consistency of some of the results. We are thus somewhat likely to accept this manuscript for publication, providing that you modify the manuscript according to the review recommendations and address the concerns of the reviewers (this *may* have to involve additional experiments if the reviewers points cannot be addressed by analysis alone). We would also appreciate if you could format the manuscript (perhaps a resubmission?) in the PLOS style when you respond, it will help a more fluent editorial and review process. 

Please prepare and submit your revised manuscript within 30 days (the standard frame for minor revision). If you anticipate any delay, please let us know the expected resubmission date by replying to this email and we will be happy to grant you substantial more time if needed.

Sincerely,

Aldo A Faisal

Associate Editor

PLOS Computational Biology

Wolfgang Einhäuser

Deputy Editor

PLOS Computational Biology

[LINK]

Reviewer's Responses to Questions

**Comments to the Authors:**

Reviewer #1: The study by Tsay and colleagues investigates how target error (TE) modulates implicit learning, which is based on sensory prediction error (SPE) during adaptation to visuomotor rotation. They used the trial-by-trial error clamp learning as the testbed and modulated the TE by “jumping” the target during the reaching movement. In this particular instantiation of both TE and SPE, they find that the hand deviation in the very next trial is modified when both SPE and TE are present, but not when only TE is present. In the former case, the target jump attenuates the implicit learning as a function of TE size. Also, the target jump with 0 TE also attenuates the learning, implying that distraction is the reason behind TE's influence. They also build several models to explain the influence of TE on SPE-based learning; the conclusion is that the target jump distracted people, thus attenuated learning.

The study consists of 4 experiments with a progressive exploration of the research questions. I find it technically sound with relatively clear data. The finding that TE alone could not drive implicit adaptation is timely in the field.

I have two major concerns: 1) the interpretation of the findings and 2) the seemingly conflicting results between experiments.

1) The TE-only conditions basically provide a task-irrelevant zero-clamp cursor, which is assumed to make SPE zero. Then, the null effect of the target jump is observed in experiments 1 and 2. Hypothetically, if there is no cursor feedback shown during reaching, i.e., removing the zero-clamp manipulation, we would probably see the effect of the target jump. In this hypothetical experiment, the SPE is arguably zero or absent, but the TE starts to have its implicit learning effect. I raise this question to scrutinize the current interpretation of the absence of learning effect with a target jump + a zero-clamp cursor. Instead of claiming the TE would not lead to implicit learning, we could also interpret the findings as the zero-clamp helps participants ignore the target jump? In other words, the unusual combination of zero-clamp and target jump, while both are instructed to be ignored, leads to the absence of responses to target jump. Whether it can be generalized to a conclusion that pure TE would not cause any implicit learning shall be questioned.

2) The discrepancies between experiments:

Why the same perturbations (say, +/-4 degree SPE+TE) produced different learning sizes in Exp1 and Exp2A, for example? See Figure 2c and 2e. Can it be attributed to the block vs. random design?

Exp2A-B and Exp 3A also differed quite a bit (figure 4). If we assume the 3-degree and 4-degree error clamp adaptation is similar (as they should be), we would expect to see a similar learning modulation of recalibration by the target jump. Thus, the modulation shall be linear, as shown in Exp3A and 3B and in the proposed Distracted SPE model. However, we observed a few obvious differences between the two experiments. For example, for a jump to -4 degree, Exp2 still produced large learning amounting to about 1 degree for an effective TE of size 0; in contrast, for a jump to -3 degree, Exp3 produced a nearly zero learning for an effective TE of size 0. How to reconcile the considerable difference between these two experiments, besides that Exp2 was performed in person and Exp3 was performed online?

Minor issues:

Issues with the description of the experimental design:

Line 124: it is unclear to me how the authors examined the learning without providing TE-only and SPE+TE trials in separate blocks. Exp1 essentially has a mini-block design.

Line462: what is the so-called random, zero-mean order? Five conditions per mini-block? Are conditions randomized across trials within a mini-block? It is unclear how the experiment trials are organized for these two experiments. Figures 2a and 2b give the impression that the trial after a perturbation (trial n+1) is a catch trial with SPE=0 and TE=0. But this is not the case, right? Then, how can we evaluate the learning in the next trial?

Line 157: less than “twice” the magnitude of the cursor perturbation??

Line166: the citation format shall be checked throughout. Here a full name is used.

Line181: In to in

Figure 4: it might be better to show the linear trend in a and c by connecting the dots representing “jump-in-place 0 deg” condition. Thus all three data points are for conditions with a jumping target.

Line241: the findings did not provide direct support for the statement that the target jump modulates the size of the TE. In fact, we assume the target jump affects TE size, and we observe that implicit recalibration is modified as a result (besides the distraction effect).

Line261: the slope of the TE function is about 0.02. Does it also hold for Exp1 where a larger target jump is used?

Line270 and Line272: the numbers are all approximate; please state it clearly.

Line 271: The real test of the model should be testing its predictions with fitted parameters. For example, the winning model, the Distracted SPE+TE model, predicts that the SPE learning shall be effectively zero with a target jump larger than 35 degrees. This is a strong prediction, which can be easily tested. Currently, the model comparisons are about model fitting without rigorous testing.

Table 2: negative R squares?

Reviewer #2: In the article “Interactions between sensory prediction error and task error during implicit motor learning,” the authors look to determine the influence/interaction between different error signals on implicit changes in movement. Participants reached to targets while the sensory prediction error (visual clamp) and task error (target jump) were manipulated. Results indicated that task errors failed to lead to implicit changes in reaches in the absence of a sensory prediction error. However, when a sensory prediction error was present, task error modulated implicit changes in reaches. As well implicit changes in reaches were modified even if the target simply flickered in place. The authors went on to look to model their results using 3 different computational models. Results were best accounted for by a computational model in which sensory prediction errors and task errors contribute to implicit changes in reaches, and this contribution is modulated by attention.

The paper poses an interesting and timely question, as currently there is much work being done to discover the processes underlying implicit and explicit motor adaptation. The inclusion of the modelling work strengthens the paper and provides direction for future work. Below I outline a few questions/comments.

The paper does not follow the typical organization of articles in PLOS, including the methods at the end of the manuscript. I assume this needs to be altered moving forward, and may make the manuscript easier to follow. As well, the reference to figures and panels in figures is not always correct (e.g., see Figure 3 caption).

The authors speak of the work providing insight into motor learning/adaptation (though the word recalibration is also used). In the current paradigm, error signals change from trial-to-trial. Thus, it is unclear if results can provide insight into motor learning, as there are no long term changes in behaviour. The authors should make clear how these results can/cannot speak to previous findings in visuomotor adaptation literature, which traditionally consider implicit adaptation as reflecting permanent changes in reach performance (i.e., aftereffects) that arise slowly.

The introduction of the target jump and the resulting target error signal is an interesting manipulation. Previous work using target jumps, has suggested that the hand is automatically drawn to the new target position unconsciously (e.g., Goodale and Milner’s dual visual systems model; Pisella and colleagues’ work on the automatic pilot). How does the potential for the hand to be automatically updated in response to a target jump (even when it is not supposed to move in the direction of the target jump) influence the current results and resulting interpretation?

The authors speak to attention/distraction as having an influence on implicit recalibration. How does the availability of the sensory prediction error influence results? For example, the stationary target jump/flicker would have decreased the time that participants experienced the sensory prediction error (potentially giving rise the fixed cost observed). Alternatively, could other factors arising as a result of change in paradigm influence performance (i.e., not just attention)?

In the current study, 2 experiments were conducted online. The general data trends look similar to the results presented for the first 2 in-person experiments. That said, I am curious what the data looks like with respect to RT, MT, variability in performance measures, etc. Given the lack of published data related to online experiments, it would be beneficial to have sample data available.

**Have the authors made all data and (if applicable) computational code underlying the findings in their manuscript fully available?**

Reviewer #1: **No: **I did not see anywhere in the paper stating the availability of the data and the model codes

Reviewer #2: **No: **Will make available upon article acceptance

PLOS authors have the option to publish the peer review history of their article (what does this mean?). If published, this will include your full peer review and any attached files.

Reviewer #1: No

Reviewer #2: No

Figure Files:

Data Requirements:

Reproducibility:

References:

---

## [Decision Letter · Decision Letter 1]

30 Jan 2022

Dear Dr. Tsay,

Thank you very much for submitting your manuscript "Interactions between sensory prediction error and task error during implicit motor learning" for consideration at PLOS Computational Biology. As with all papers reviewed by the journal, your manuscript was reviewed by members of the editorial board and by several independent reviewers. The reviewers appreciated the attention to an important topic. Based on the reviews, we are likely to accept this manuscript for publication, providing that you modify the manuscript according to the review recommendations. To be clear, these are really very minor revisions that we request, especially addressing the extra detail for the new experiment.

Sincerely,

Aldo A Faisal

Associate Editor

PLOS Computational Biology

Wolfgang Einhäuser

Deputy Editor

PLOS Computational Biology

[LINK]

Reviewer's Responses to Questions

**Comments to the Authors:**

Reviewer #1: The revision has successfully addressed most concerns that I raised. In particular, the newly added experiment with a random TE but without SPE is a nice add-on for supporting the conclusions.

However, this new experiment is not described in the revision; only a result figure is presented as Fig S1 (note the figure is not in the redlined manuscript either). As this experiment is critical for the main conclusion, I suggest including it in the main text. As I do not know the details of the experiment, I assume that the participants were instructed to ignore the random target jumps. Did they?

With this new experiment, it becomes clear that we need to interpret the findings more cautiously than now. The current conclusion is strong, i.e., target error alone cannot support implicit recalibration/learning. But I would suggest concluding this effect with its associated conditions.

1) Are the target jump findings generalizable to all learning based on target error? Saccadic adaptation is a similar implicit adaptation paradigm with a target jump in the middle of the movement, leading to trial-by-trial adaptation. Instead of ignoring the target jump, the instruction for saccadic adaptation is to make saccades to the target. How can we reconcile the discrepancy between the present study and saccadic adaptation? Can we explain it away by saying saccadic adaptation is not implicit, or their target error is task-contingent, but here it is not?

2)The zero slope observed in the new experiment is based on a fitting over a wide range of target errors (-16 to 16 degrees). Interestingly, there is a small negative slope if only the three small errors are included (-4, 0, and 4 degrees). If we assume the 16-degree jump is more explicit, but the 4-degree jump is more implicit, it appears that the target jump can lead to “pure” implicit learning. Of course, I would love to see the regression results for these small jumps.

3)what if the target jump is not presented in a fully random fashion but, say, in mini-blocks? If it is purely random, the nervous system might be better off aiming for the middle and thus completely ignoring the target error. That’s being said, should the current conclusion about zero effect of TE on implicit learning be limited to random trial-by-trial learning? Relatedly, the discussion about the random vs. block is not precise in Discussion (Line321). None of the referenced studies support the conclusion that “TE in the absence of SPE failed to elicit implicit recalibration” with a block design. This is still an unexplored issue.

Line699: the reference is not in the right format. Please check other places too.

Reviewer #2: I thank the authors for their careful consideration of the comments/suggestions raised in the first round of reviews. The paper has been greatly improved. I have no further comments at this time.

**Have the authors made all data and (if applicable) computational code underlying the findings in their manuscript fully available?**

Reviewer #1: **No: **they said the data will be available by request, but the model codes are not shared.

Reviewer #2: Yes

PLOS authors have the option to publish the peer review history of their article (what does this mean?). If published, this will include your full peer review and any attached files.

Reviewer #1: No

Reviewer #2: No

Figure Files:

Data Requirements:

Reproducibility:

References:

---

## [Decision Letter · Decision Letter 2]

9 Mar 2022

Dear Dr. Tsay,

We are pleased to inform you that your manuscript 'Interactions between sensory prediction error and task error during implicit motor learning' has been provisionally accepted for publication in PLOS Computational Biology.

Best regards,

Aldo A Faisal

Associate Editor

PLOS Computational Biology

Wolfgang Einhäuser

Deputy Editor

PLOS Computational Biology

Reviewer's Responses to Questions

**Comments to the Authors:**

Reviewer #1: The revision has successfully addressed all my concerns. The new analyses and the mentioning of some unpublished datasets make the argument of the paper more convincing. well done!

Reviewer #2: Thank you for taking time to address the requested revisions (including additional details regarding Experiment 1B in the main text). I have no further comments/suggestions at this time.

**Have the authors made all data and (if applicable) computational code underlying the findings in their manuscript fully available?**

Reviewer #1: Yes

Reviewer #2: Yes

PLOS authors have the option to publish the peer review history of their article (what does this mean?). If published, this will include your full peer review and any attached files.

Reviewer #1: No

Reviewer #2: No

---

## [Editor Report · Acceptance letter]

18 Mar 2022

PCOMPBIOL-D-21-01291R2 

Interactions between sensory prediction error and task error during implicit motor learning

Dear Dr Tsay,

I am pleased to inform you that your manuscript has been formally accepted for publication in PLOS Computational Biology. Your manuscript is now with our production department and you will be notified of the publication date in due course.

With kind regards,

Zsofia Freund
